# The Role of Triazole and Glucose Moieties in Alkali Metal Cation Complexation by Lower-Rim Tertiary-Amide Calix[4]arene Derivatives

**DOI:** 10.3390/molecules27020470

**Published:** 2022-01-12

**Authors:** Josip Požar, Marija Cvetnić, Andrea Usenik, Nikola Cindro, Gordan Horvat, Katarina Leko, Matija Modrušan, Vladislav Tomišić

**Affiliations:** Department of Chemistry, Faculty of Science, University of Zagreb, Horvatovac 102a, 10000 Zagreb, Croatia; marija.cvetnic@chem.pmf.hr (M.C.); ausenik@chem.pmf.hr (A.U.); ncindro@chem.pmf.hr (N.C.); ghorvat@chem.pmf.hr (G.H.); kleko@chem.pmf.hr (K.L.); mmodrusan@chem.pmf.hr (M.M.)

**Keywords:** calix[4]arenes, alkali metal cations, complexation, solvation, calorimetry, molecular dynamics

## Abstract

The binding of alkali metal cations with two tertiary-amide lower-rim calix[4]arenes was studied in methanol, *N,N*-dimethylformamide, and acetonitrile in order to explore the role of triazole and glucose functionalities in the coordination reactions. The standard thermodynamic complexation parameters were determined microcalorimetrically and spectrophotometrically. On the basis of receptor dissolution enthalpies and the literature data, the enthalpies for transfer of reactants and products between the solvents were calculated. The solvent inclusion within a calixarene hydrophobic *basket* was explored by means of ^1^H NMR spectroscopy. Classical molecular dynamics of the calixarene ligands and their complexes were carried out as well. The affinity of receptors for cations in methanol and *N,N*-dimethylformamide was quite similar, irrespective of whether they contained glucose subunits or not. This indicated that sugar moieties did not participate or influence the cation binding. All studied reactions were enthalpically controlled. The peak affinity of receptors for sodium cation was noticed in all complexation media. The complex stabilities were the highest in acetonitrile, followed by methanol and *N,N*-dimethylformamide. The solubilities of receptors were greatly affected by the presence of sugar subunits. The medium effect on the affinities of calixarene derivatives towards cations was thoroughly discussed regarding the structural properties and solvation abilities of the investigated solvents.

## 1. Introduction

One of the main goals in supramolecular chemistry is the preparation of efficient, selective, and possibly water-soluble receptors. These requirements are difficult to meet in a single chemical species. Namely, a well-defined and solvent-shielded binding site can be achieved by incorporating rigid nonpolar functionalities into the host backbone, which, however, always results in its poor solubility in water. The calixarenes are a class of macrocyclic compounds that can serve as a proof of point [1,2,3,4,5,6]. Their stiff frame and the possibility of both upper- and lower-rim functionalization have led to preparation of exceptional ligands for charged and neutral species in a range of media but very rarely in water [7,8,9,10,11,12,13,14,15,16]. With the aim of overcoming this obstacle, we have recently introduced a new class of water-soluble glycocalixarenes designed for efficient hosting of first-group cations [16]. These were lower-rim derivatives possessing tertiary amide groups and triazole-sugar subunits (Figure 1, compound **l**). As is customary in calixarene chemistry, the *cone* conformation was assured by bulky *tert*-butyl groups situated at the upper rim. The affinity of glycocalixarene **l** towards Na^+^ was beyond expectation. The complex stabilities with **l** in water were very similar to those of better preorganized azacryptands [17] and much larger than those with crown ethers [18].

It should be noted that several water-soluble calixarenes have already been described in the literature. In order to assure their solubility in water, charged functionalities or those undergoing (de)protonation reactions in water were introduced both at the upper and lower rim. Unfortunately, such calixarenes exhibit pH-dependent receptor properties, form complexes with counterions, and unselectively bind a variety of charged species [19,20,21]. A series of neutral water-soluble glycocalixarenes was used for cell recognition and binding [22]. However, sugar subunits were never introduced solely to assure the compound solubility in water, whereby the macrocycle also contained functionalities for recognition of particular guests. A simple click-coupling via triazole subunits provides a path for synthesis of calixarene-based receptors for a variety of chemical species in water [23,24,25,26,27]. This is important in the design of selective ligands for metal cations [16], calixarene-based biomimetic compounds [12,28,29], and supramolecular systems, which could be potentially used as biologically active species [30,31]. On the other hand, the role of triazole subunits in alkali metal cation recognition by corresponding lower-rim tertiary amide glycocalixarenes was not previously investigated. However, by comparing the affinities of glycosylated calixarene with its peracetylated precursor, it was established that the glucose subunits do not participate in cation binding [16]. This does not necessarily imply that they do not influence the binding via intramolecular hydrogen bonding or preorganization of the host binding site. With the aim of elucidating the roles of triazole and glucose moieties in the alkali metal cation complexation reactions, we have synthetized a triazole calixarene derivative (**L, Figure 1**a) and compared the complex stability constants and other thermodynamic reaction parameters with those corresponding to previously prepared glycocalix[4]arene **l** (Figure 1b) [16]. The binding reactions of compound **L** were studied in *N,N*-dimethylformamide (DMF), methanol (MeOH), and acetonitrile (MeCN). First, two solvents were chosen for the purpose of comparison with glycoconjugate analogue **l**, as both ligands exhibit sufficient solubility required for complexation investigations in these media. The thermodynamic parameters of alkali metal cation reactions with **l** in *N,N*-dimethylformamide and methanol are reported herein, whereas the stability constants of the complexes in the latter solvent have been published as a part of our earlier study [16]. The solvent effect on the complexation reactions with **L** and **l** was particularly addressed, both in the context of reactants and product solvation. To gain a more detailed insight into the investigated reactions and the structures of free and complexed ligands, the classical molecular dynamics simulations were carried out as well.

## 2. Results and Discussion

### 2.1. Complexation of Alkali Metal Cations with ***L*** in Methanol

The alkali metal cation binding by compound **L** in MeOH was studied calorimetrically. As an example of the results obtained, the microcalorimetric titration of receptor with LiClO_4_ is shown in Figure 1. The cation binding was accompanied with negative enthalpy changes. The complex stability constant and the reaction enthalpy were processed according to a 1:1 binding model. The standard reaction Gibbs energy and entropy were calculated using fundamental thermodynamic relations. As seen from the data listed in Table 1, the receptor **L** is a modest binder of Li^+^ cation. This is due to relatively low reaction enthalpy (absolute value) and almost negligible entropy.

Contrary to the moderate affinity of calixarene **L** for Li^+^, the complexation of Na^+^ (Appendix A) was far more thermodynamically favorable. The recorded successive enthalpy changes did not depend on the cation-to-ligand ratio up to equivalence, which confirmed 1:1 binding stoichiometry and indicated that lg ***K***(Na**L**^+^) > 6. The sodium complex stability constant was, therefore, determined by carrying out microcalorimetric displacement titrations, i.e., the experiments in which the complexed K^+^ was expelled by sodium cation (Figure 2 and Table 1).

This required prior determination of K**L**^+^ stability constant, which was obtained by direct microcalorimetric titrations (Table 1, Appendix A). The binding of larger alkali metal cations can be characterized as weak. The stability of Rb**L**^+^ was considerably lower than that of Li**L**^+^ (Table 1, Appendix A), whereas the Cs^+^ binding, though observed calorimetrically (Appendix A), was too weak for reliable determination of the corresponding thermodynamic complexation parameters. This was confirmed by conducting spectrophotometric titrations of receptor with CsCl up to *N*(Cs^+^)/*N*(**L**) = 350 (Appendix A).

By examining all standard thermodynamic complexation parameters in MeOH (Table 1), it can be clearly concluded that the reactions are enthalpy driven. The Δ_r_*S*° was strongly unfavorable for all complexation reactions, apart from Li^+^ coordination. Due to the pronounced orientation of alcohol molecule dipoles around high-charge-density Li^+^, its desolvation is particularly entropically beneficial and enthalpically demanding [32], which is clearly reflected in the corresponding Δ_r_*H*° and Δ_r_*S*° values. As already mentioned, the peak affinity was observed for Na^+^, which is best suited to the size of the binding site [3,10,15,33]. In accordance, its complexation was the most enthalpically favored. Interestingly, much lower stability of Rb**L**^+^ compared to K**L**^+^ is almost entirely due to the differences in Δ_r_*S*°. This might be indicative of an entropy loss as a consequence of partial desolvation of bound Rb^+^, which was observed during the corresponding MD simulations (Section 2.7).

### 2.2. Complexation of Alkali Metal Cations with ***L*** in Acetonitrile

Acetonitrile was a far more suitable complexation medium compared to methanol. The receptor **L** exhibited such high affinity for cations smaller than Rb^+^ (Figure 3, Table 1) that the stability constants of the corresponding complexes had to be determined by a series of displacement titration experiments. This was accomplished as follows: standard thermodynamic reaction parameters for Rb^+^ binding were determined from direct calorimetric titration experiments (Figure 4a, Table 1). The stability constant of K**L**^+^ was obtained by processing the enthalpy changes due to displacement of Rb^+^ from the ligand binding site, that of Na**L**^+^ by analogous displacement of bound K^+^, and that of Li**L**^+^ by Li^+^ displacement with Na^+^ (Appendix A, Figure 4b, Appendix A, and Table 1). The complexation reaction enthalpies were determined from direct calorimetric titrations (Appendix A; Table 1). The complexation of Cs^+^ was studied by direct calorimetric titrations (Appendix A, Table 1). The enthalpy changes of CsI(MeCN) dilution were rather high. The reliability of the calorimetrically determined stability constant of Cs**L**^+^ was, therefore, checked by performing UV–Vis spectrophotometric titrations (Appendix A), which could be carried out up to higher cation-to-ligand molar ratios. The agreement between the values obtained by both methods is very good (Table 1).

By comparing the thermodynamic data for cation complexation in MeOH and MeCN (Table 1), one can notice that the reaction entropies are far less unfavorable in the latter solvent. The complexation is more exothermic in MeCN, except for Rb^+^, for which the Δ_r_*H*° is similar in both solvents. Remarkably, such relation of Δ_r_*H*° and Δ_r_*S*° values led to more than seven orders of magnitude higher stability constant of Li**L**^+^ in MeCN compared to MeOH (Figure 3, Table 1). As the cation radius increases, the differences between the complex stability in explored solvents become less pronounced. This is almost certainly in part due to more favorable solvation of potassium and rubidium cations in acetonitrile, while the opposite holds for alkali metal cations of a smaller radius [32].

However, when discussing the solvent effect on the complexation equilibria, the receptor and the product solvation in MeOH and MeCN have to be taken into account. For that purpose, the standard thermodynamic transfer parameters for compound **L** and the corresponding alkali metal cation complexes among the solvents should be determined. These values can be calculated from the receptor solubilities and dissolution enthalpies, as well as the standard thermodynamic transfer functions of free cations available in literature (Section 2.6).

### 2.3. Complexation of Alkali Metal Cations with ***L*** in N,N-Dimethylformamide

The hosting of alkali metal cations with receptor **L** was also investigated in *N,N*-dimethylformamide (Appendix A). The determined thermodynamic complexation parameters are listed in Table 1.

The complex stabilities are lower than in MeCN and MeOH (Figure 3, Table 1). This is not surprising considering the strongly favorable solvation of alkali metal cations in DMF. In fact, based solely on the Gibbs energy of their transfer from MeCN and MeOH to DMF (Δ_t_*G*°(Li^+^(MeCN) → Li^+^(DMF) = −50.9 kJ·mol^−1^; Δ_t_*G*°(Li^+^(MeOH) → Li^+^(DMF) = −25.4 kJ·mol^−1^) [32], the complexation in the latter is not even likely to be observed. The corresponding Δ_t_*G*° for all other alkali metal cations are not as strongly exergonic, however, indicate that considerably lower affinities for alkali metal cations in DMF are expected. On the other hand, DMF is, compared to MeOH and MeCN, a particularly favorable medium for solvation of the complexed cations as well [15,34], which is favorable for the hosting process. As in MeOH and MeCN, the compatibility of Na^+^ and the ligand **L** binding site sizes leads to the most exothermic complexation and the largest stability of Na**L**^+^ species in DMF (Table 1). As can be clearly seen in Figure 3, the affinity of **L** for K^+^ is similar to that for Li^+^, whereas the binding of larger Rb^+^ and Cs^+^ was not observed. All complexation reactions are enthalpically driven, whereby the Δ_r_*H*° values for the first group cation hosting were the least favorable among the studied solvents. The only exception is Li^+^, whose complexation is more exothermic compared to MeOH. That is not in accord with thermodynamically more favorable solvation of Li^+^ in DMF. This finding can again serve as a clear indication that, apart from cation solvation, that of the complex and the receptor should be considered when discussing the solvent effect on the complexation equilibria (Section 2.6).

### 2.4. Complexation of Alkali Metal Cations with Compound ***l*** in Methanol and N,N-Dimethylformamide

Thermodynamic parameters of reactions involving alkali metal cations and receptor **l** in methanol were determined microcalorimetrically (Appendix A). As seen from the data presented in Table 2, the affinity of glycocalixarene for Li^+^ is moderate, which is a consequence of relatively low reaction enthalpy (absolute value) and almost negligible entropy (Table 2). This was also noticed in the case of the corresponding reaction with compound **L**. Moreover, Δ_r_*H*° and Δ_r_*S*° for Li^+^ binding with both ligands were almost the same. The hosting of Na^+^ by calixarene **l** was examined in our earlier study [16]. The obtained Δ_r_*H*° and Δ_r_*S*° values amounted to −59.1 kJ·mol^−1^ and −59.7 J·K^−1^ mol^−1^ (Table 2), which is highly similar to the case of sodium cation complexation with compound **L** (Table 1 and Appendix A).

As can be seen in Table 2, the stability of K**l**^+^ was notably lower compared to Na**l**^+^, which is almost entirely due to the less favorable complexation enthalpy. The rubidium hosting is strongly entropically disadvantageous, leading to the relatively low affinity of the glycocalixarene for this cation. Again, the values obtained for complexation of potassium and rubidium cations with compounds **L** and **l** are very similar. The titration of receptor **l** with Cs^+^ in methanol did not result in measurable enthalpy changes, indicating the low affinity of glycocalixarene for the largest alkali metal cation [16]. This is in agreement with previously reported spectrophotometric investigations. Besides that, the spectrophotometrically determined complex stability constants [16] are very similar to the herein obtained values.

The standard thermodynamic reaction parameters for complexation of glycocalixarene **l** in DMF are also given in Table 2. The corresponding thermograms, and experimental and calculated enthalpy changes as a function of cation-to-host molar ratio are shown in Appendix A. The stability constants of alkali metal cation complexes with glycocalixarene **l** are highly similar to those with the triazole derivative **L** (Table 1 and Appendix A). As in methanol, this is a consequence of relatively small differences among Δ_r_*H*° and Δ_r_*S*° values for each M^+^–ligand pair.

The comparative study of **l** and **L** binding affinities for alkali metal cations in DMF and MeOH, therefore, indicates that the presence of glucose subunits has very little effect on their hosting properties. However, as stated in the Introduction, it has a profound effect on the calixarene solubility. This qualitative observation can, from the thermodynamic point of view, be rationalized by means of standard transfer functions of receptors **l**, **L**, and glucose among the reaction media of interest, which are discussed in the following section.

### 2.5. The Solvation of Receptors in Studied Solvents

The solubilities of compounds **L** and **l** in MeOH and DMF were investigated with the aim of determining the standard Gibbs energies of their transfers using the relation [3,35]:(1)$$\Delta_{t}G{}^{\circ}(\mathrm{host}(\mathrm{MeOH})\to \mathrm{host}(\mathrm{DMF}))=RT\;\ln\left(\frac{s_{\mathrm{MeOH}}}{s_{\mathrm{DMF}}}\right),$$
where *s* denotes solubility of the compound. The relation is valid if the solubility of the compound is low (activity coefficients can then be approximately equal to unity) and if there is no transformation of solid into a solvate [36]. The solubilities of ligand **L** in MeOH and DMF were too high (*s* > 0.1 mol·dm^−3^) for the first assumption to be valid.

In contrast to compound **L**, the solubility of calixarene **l** was quite low in acetonitrile (< 1 × 10^−5^ mol·dm^−3^), low in MeOH (*s* = 3.69 × 10^−3^ mol·dm^−3^), and quite high in DMF (*s* > 0.1 mol·dm^−3^). Consequently, the standard Gibbs energies of transfers for compounds **L** or **l** among any pair of listed solvents could not be determined. On the other hand, the enthalpies of transfer (Δ_t_*H*°) were obtained calorimetrically according to expression:(2)$$\Delta_{t}H{}^{\circ}\left(L\left(\mathrm{MeOH}\right)\to L\left(\mathrm{DMF}\right)\right)=\Delta_{s}H{}^{\circ}\left(\mathrm{DMF}\right)-\Delta_{s}H{}^{\circ}\left(\mathrm{MeOH}\right),$$
where Δ_s_*H*° denotes the ligand **L** dissolution enthalpy in a particular solvent, in this case, MeOH or DMF. The analogous equation was used for compound **l**. The thermograms for dissolution of calixarenes **L**, **l**, and glucose (Glc) in the investigated solvents are shown in Appendix A. The obtained Δ_s_*H*° and Δ_t_*H*° for both compounds are listed in Table 3.

The enthalpy of transfer of compound **L** from methanol to *N,N*-dimethylformamide is quite low: Δ_t_*H*°(**L**(MeOH) → **L**(DMF)) = −2.3 kJ·mol^−1^. The receptor **L**, hence, only weakly prefers the interactions with DMF. The transfer of calixarene **L** from methanol to acetonitrile is endothermic: Δ_t_*H*°(**L**(MeOH) → **L**(MeCN)) = 7.8 kJ·mol^−1^, indicating the least favorable solvation energetics in the latter solvent.

The differences in receptor transfer enthalpies are remarkable. The Δ_t_*H*°(**l**(MeOH) → **l**(DMF)) is approximately 80 kJ·mol^−1^ more favorable compared to the corresponding value for compound **L**. This is obviously a consequence of structural differences among the ligands, predominantly related to the fact of whether they contain glucose subunits or not. Namely, the transfer of free glucose among the mentioned solvents is considerably exothermic (Δ_t_*H*°(Glc(MeOH) → Glc(DMF)) = −7.9 kJ·mol^−1^; Table 3). Besides, when the transfer enthalpy corresponding to a process of eight free glucose molecules and one ligand **L** molecule transfer from MeOH to DMF is calculated:(3)L(MeOH)+8 Glc(MeOH)→L(DMF)+8 Glc(DMF).

A value of −65.5 kJ·mol^−1^ is obtained (Table 3). This indicates that, enthalpy-wise, the transfer of glycocalixarene **l** among these solvents can be roughly approximated with a transfer of compound **L** and eight free glucose units. The differences between the mentioned transfer enthalpies arise from the fact that the solvations of free and bound glucose are expected to occur via different solvation patterns (number of DMF molecules and their arrangement). Compound **L** also contains additional ethyl groups, whereas one OH group in free glucose becomes coupled to a triazole ring via ether bond in receptor **l**.

The strong preference of compound **l** to engage in interactions with DMF and MeOH compared to MeCN (***s***(**l**, DMF) > ***s***(**l**, MeOH) > ***s***(**l**, MeCN)) can be explained by the differences in their Lewis basicities, which are quantified by Gutmann donor numbers (DN) [34]: DN(DMF) = 111 kJ·mol^−1^, DN(MeOH) = 79 kJ·mol^−1^, DN(MeCN) = 59 kJ·mol^−1^ [37,38]. The larger DN of the solvent should lead to stronger hydrogen bonds with OH groups, which seems to be particularly important for solvation of glucose subunits in receptor **l**.

Apart from complexation studies, the affinity of **L** and Na**L**^+^ for inclusion of MeOH, MeCN, and DMF was explored in CDCl_3_ by means of ^1^H NMR spectroscopy. The addition of solvents to solution of **L** did not result in changes in the corresponding NMR spectra, whereas shifts of Na**L**^+^ aryl and *tert*-butyl protons upon MeOH or MeCN addition were observed (Figure 5 and Appendix A). The NMR spectrum of Na**L**^+^(CDCl_3_) remained unaltered upon titration with DMF (Appendix A). The described results indicated the formation of Na**L**^+^ adducts with MeOH and MeCN, whose stability constants (lg [*K*(Na**L**MeCN^+^)/dm^3^ mol^−1^] = 1.73, lg [*K*(Na**L**MeOH^+^)/dm^3^ mol^−1^] = 0.83) were determined by nonlinear regression analysis of NMR data. In accord with the literature data [15], the sodium complex preferred acetonitrile over methanol, which can be explained by favorable interaction of acetonitrile protons with the electron-rich cavity of the receptor. The difference between the **L** and Na**L**^+^ affinities for solvent molecules is due to the larger conformational freedom of the free receptor [7,9,10,11,15,39]. The results of the herein described computational experiments (Section 2.7) are in agreement with this experimental data.

### 2.6. The Solvent Effect on the Alkali Metal Complexation and Comparison of ***L*** and ***l*** Binding Affinities

The difference in any standard thermodynamic reaction quantity (Δ_r_*X°, X = G, H, S*) among the media of interest (e.g., DMF and MeOH) can be expressed as follows:(4)$$\begin{array}{ll} \Delta_{r}X{}^{\circ}\left(\mathrm{DMF}\right)-\Delta_{r}X{}^{\circ}\left(\mathrm{MeOH}\right) & =\Delta_{t}X{}^{\circ}\left(ML^{+},\mathrm{MeOH}\to \mathrm{DMF}\right)\\ & -\Delta_{t}X{}^{\circ}\left(M^{+},\mathrm{MeOH}\to \mathrm{DMF}\right)\\ & -\Delta_{t}X{}^{\circ}\left(L,\mathrm{MeOH}\to \mathrm{DMF}\right), \end{array}$$
where Δ_t_*X*° represents the standard thermodynamic parameter for transfer of reactants (M**^+^**, **L**) and product (M**L^+^**) [3,11,15,35]. The Δ_t_*X*° of free cations from MeOH to any other solvent (e.g., DMF) were obtained by combining the functions of transfer from water: (5)$$\Delta_{t}X{}^{\circ}\left(M^{+},\mathrm{MeOH}\to \mathrm{DMF}\right)=\Delta_{t}X{}^{\circ}\left(M^{+},H_{2}O\to \mathrm{DMF}\right)-\Delta_{t}X{}^{\circ}\left(M^{+},H_{2}O\to \mathrm{MeOH}\right).$$

The data, based on Ph_4_AsPh_4_B convention, were taken from [32]. Such analysis for Na^+^ complexation resulted in the thermodynamic cycle presented in Figure 2, which explains the differences in standard reaction enthalpies for the binding of Na^+^ in methanol (chosen as a reference solvent) and *N,N*-dimethylformamide.

As can be seen, the energetics of ligand solvation are slightly more favorable in DMF, whereas the transfer of Na^+^ from methanol to *N,N*-dimethylformamide is considerably exothermic. From the point of view of reactant interactions with the solvents, methanol is favored as complexation media. The difference between the complexation enthalpies in two media is notably influenced by the solvation of the products, which is more favorable in DMF. This solvent is, due to its large dipole moment and Gutmann donor number [34,37,38], a particularly favorable media for both free and complexed cation solvation [15]. As a result, the Δ_r_*H*°(DMF)–Δ_r_*H*°(MeOH) is lower than would be expected by considering only the energetics of reactant solvation. The corresponding schemes for the reactions of **L** with other alkali metal cations are given in the Appendix A. In the case of Li^+^ complexation, the energetically more favorable reaction in DMF is mostly due to the exothermic transfer of the complex. In contrast, the more negative K^+^ reaction enthalpy in MeOH is predominantly a consequence of strongly favorable cation solvation in DMF.

The thermodynamic cycle for Na^+^ complexation accounting for the differences in standard reaction enthalpy in methanol (reference solvent) and acetonitrile is shown in Figure 3. The enthalpy of complex transfer from MeOH to MeCN is close to zero. The more exothermic complexation in acetonitrile is, hence, primarily a consequence of stronger interactions of both reactants with methanol, which favors acetonitrile as a complexation medium. The corresponding schemes for the reactions of other alkali metal cations are given in the Appendix A. According to the thermodynamic cycle shown in Appendix A, the free Li^+^ cation strongly prefers interactions with MeOH, whereas the opposite holds for the corresponding complex, leading to a much more exothermic complexation in acetonitrile. The transfers of K^+^ and Rb^+^ from MeOH to MeCN are both exothermic. This fact, the enthalpically favorable transfer of K**L**^+^, and the almost isoenthalpic transfer of Rb**L**^+^ lead to relatively small difference in complexation enthalpies in the investigated solvents for Rb^+^ and to a significant enthalpical preference of formation of K**L**^+^ in MeCN.

The solvent influence on the energetics of alkali metal cations complexation with receptor **l** has been analyzed analogously (Figure 4, Appendix A). As already stated, the transfers of free cations from MeOH to DMF are considerably exothermic, however, not nearly as so as those of free ligand and the complexes. Interestingly, the difference between the latter transfer enthalpies amounts to 9.3 kJ·mol^−1^ at most, so they largely cancel each other out in their contribution to Δ_r_*H*°(DMF)–Δ_r_*H*°(MeOH). This is reasonable, taking into account the established weak influence of glucose subunits on the cation hosting and extremely favorable interactions of these subunits with DMF.

Namely, the differences in standard complexation Gibbs energies Δ(Δ_r_*G*°) for receptor **l** and **L** in methanol and *N,N*-dimethylformamide are almost within the experimental error (Appendix A). On the other hand, the transfer of **l** and glucose from MeOH to DMF is strongly exothermic and that of receptor **L** almost isoenthalpic (Table 3).

### 2.7. Molecular Dynamics Simulations

#### 2.7.1. MD Investigations of Receptors

The results of simulations of free receptors in acetonitrile, methanol, and *N,N*-dimethylformamide suggest that **L** accommodates MeCN and MeOH solvent molecules in its hydrophobic cavity, whereas the inclusion of DMF was not observed (Appendix A). The inclusion of methanol was far more pronounced (Figure 6). The MD simulations of receptor **l** were conducted in methanol and *N,N*-dimethylformamide. During the simulation in methanol, the calixarene *cone* was almost always occupied by one of the solvent molecules (five different molecules were exchanged during the simulation time (Appendix A)). In *N,N*-dimethylformamide, the **l**DMF adduct was detected only during 9 % of the simulation time (Appendix A).

The conformation of **L** and **l** *baskets* resembled a squashed *cone* shape of *c*_2_ symmetry, while, upon the inclusion of solvent molecules, the shape of the *basket* changed to the almost *c*_4_ symmetrical regular *cone* (Appendix A).

#### 2.7.2. MD Investigations of Alkali Metal Cation Complexes with **L**

The MD simulations of alkali metal cation complexes in MeCN indicate far more pronounced inclusion of the solvent molecules compared to free receptor (the solvent molecule was present within the complex cavity during the whole simulation time, Figure 6). This is in line with the experimental findings (Section 2.5, Figure 5). In the case of sodium, potassium, and rubidium complexes of **L**, only MeCN adducts with the orientation of the acetonitrile methyl group towards the alkali metal cation were observed (Appendix A). Only one solvent molecule was found to occupy the Na**L**^+^ and Rb**L**^+^ complexes, while, in the case of K**L**^+^ complex, three acetonitrile molecules were exchanged in the calixarene *cone* during simulation. The shape of the calixarene *basket* of metal–cation complexes of **L** in acetonitrile resembled a regular *cone* (Appendix A), which is a result of both the cation complexation and the inclusion of acetonitrile. In the case of lithium complex, an additional adduct is formed as Li**L**MeCN’^+^ (Appendix A), in which the nitrile group of acetonitrile coordinated the metal cation, was present during 5 % of the simulation time. The solvent coordination led to the shortening of the distance between Li^+^ and the geometric center of phenol oxygen atoms (Appendix A). The interaction energy of cation–MeCN_included_ in Li**L**MeCN’^+^ was 47 kJ·mol^−1^ larger than in the predominant Li**L**MeCN^+^ adduct (methyl group oriented towards the cation, Appendix A). The former binding mode was similar to the already described coordination of calixarene-bound lithium cation by a nitrile group of included benzonitrile molecule, which has been observed both in the solid state and in structures obtained by MD simulations [11].

The time-averaged coordination percentage and the average number of cation-coordinating lower-rim donor atoms and groups are presented in Figure 7. The distributions of metal cation–carbonyl oxygen distances are given in Appendix A. The metal ions are bound by almost all ether oxygen atoms (≥ 3.7), followed by carbonyl groups, for which the coordination number increases from Li^+^ to K^+^. Smaller lithium cation is coordinated by about two carbonyl oxygen atoms in Li**L**MeCN^+^ complex, and by only one in Li**L**MeCN’^+^ (cation coordination by nitrile group of the included MeCN molecule). The average value of coordinated carbonyl oxygen atoms in Rb**L**MeCN^+^ adduct (2.3) was lower compared to Na**L**MeCN^+^ (2.9) and K**L**MeCN^+^ (3.1). The less pronounced involvement of C=O in cation coordination was, at least partially, compensated by the binding of 0.24 N2 triazole atoms (the only case of cation coordination by triazole rings of **L**).

Information on structural characteristics of **L** in methanol were also obtained by molecular dynamics simulations. The lithium, sodium, potassium, and rubidium complexes in MeOH hosted solvent molecules in their *basket* during most of the simulation time (Figure 6, Figure 8, and Appendix A). Two types of methanol adduct were observed. The more dominant form was M**L**MeOH^+^, in which the oxygen atom of methanol molecule was oriented towards the bulk of the solution (Figure 8a,c–f). The other form, in which the methanol oxygen atom coordinated the cation, was present only in the lithium complex Li**L**MeOH’^+^ (13 % of the simulation time, Figure 6 and Figure 8b, and Appendix A).

The cation coordination by MeOH in Li**L**MeOH’^+^ species led to its weaker interaction with **L**. The exchange of methanol molecules within the cavity was more prominent for complexes with smaller cations (Figure 6). In the case of the rubidium complex, the partial desolvation of rubidium cation was observed during 90 % of the simulation time where one methanol molecule was coordinated to the cation at the lower-rim side (Figure 8f). This finding is in accord with highly negative standard reaction entropy for Rb^+^ hosting (Section 2.1, Table 1). Namely, the partial Rb^+^ desolvation is expected to be less entropically favorable compared to complete elimination of MeOH from the solvation sphere.

Interestingly, while a single acetonitrile molecule was found to be included in Na**L**^+^ (Figure 6, Appendix A), 15 different MeOH molecules were exchanged and were present therein for 95 % of the simulation time (Appendix A). These findings could serve as an indication of the experimentally observed more favorable inclusion of MeCN molecule compared to MeOH (Section 2.5).

Alkali-metal cations in MeOH were coordinated by **L** through all ether oxygen atoms and some of the carbonyl groups (Figure 8, Appendix A). Only one carbonyl oxygen atom coordinated metal cation in lithium complex, whereas other cations were coordinated by 2.5 to 3 of these atoms. A rare event of coordination of triazole ring nitrogen atoms was observed for Rb^+^ complex. The average coordination number of N2 and N3 atoms was 0.03 and 0.04, respectively, which is about six times lower than that observed in acetonitrile. Distributions of metal–cation binding site atoms and cation binding site angle distributions in methanol (Appendix A) were similar to those obtained by MD simulations in acetonitrile.

The MD simulations of alkali metal cation complexes with **L** in *N,N*-dimethylformamide indicated the inclusion of DMF molecule in the hydrophobic cavity of calixarene (Appendix A). This process resulted in two types of adducts, the more dominant form being that in which the *trans* methyl group (regarding DMF oxygen atom) occupied the hydrophobic cavity (Appendix A), and the second one in which the *cis* methyl was present in the calixarene *cone* (Appendix A; these adducts are denoted with a ' symbol). The first binding mode was slightly more energetically favorable with respect to the interaction between the calixarene ligand and included DMF molecule. Internal reorientation of the included DMF molecule was observed throughout the simulations of all complexes. The adduct formation was more pronounced for lithium and sodium complexes, whereas the complexes with potassium and rubidium cations existed in the *free-basket* form during a significant portion of simulation time. The exchange of the included DMF molecules was slow on the MD timescale and more pronounced for the larger cations (from 1 to 10 molecules exchanged in the course of each simulation).

The cation coordination sphere of **L** in *N,N*-dimethylformamide included all ether oxygen atoms and a variable number of carbonyl oxygen atoms (Figure 7, Appendix A). The weak involvement of triazole nitrogen atoms in coordination was observed solely in the case of Rb**L**^+^. The distributions of metal cation–carbonyl oxygen distances and metal cation–carbonyl oxygen–carbonyl carbon angles are given in Appendix A. As is the case with other solvents, these distributions are mostly bimodal. Peaks corresponding to the bound carbonyl groups are centered at shorter distances and at angles around 110°.

#### 2.7.3. MD Investigations of Alkali Metal Cation Complexes with **l**

The inclusion of methanol molecules in the *basket* of compound **l** was observed for all studied complexes (Appendix A, Figure 9, and Appendix A). The adducts were present during most of the simulation time. The calixarene *basket* of the sodium, potassium, and rubidium complexes with included methanol molecule resembled the regular *cone*, whereas hydrophobic cavities of other complexes slightly deviated from that shape. As was the case with lithium complex of **L** in methanol, the Li**l**^+^ complex formed two types of adducts with methanol molecules, namely Li**l**MeOH^+^ and Li**l**MeOH’^+^ (Figure 9a,b). Cations were again predominantly coordinated by ether oxygen atoms, followed by carbonyl groups (Appendix A). A substantial coordination of K^+^ and Rb^+^ cations by triazole nitrogen atoms was observed, with the average coordination numbers of N2 and N3 atoms ranging from 0.2 to 0.95. The interactions of glucose units with the metal cations were not observed, which is in line with similar binding affinities of **L** and **l** for alkali metal cations in this solvent.

Distributions of metal cation binding site atoms and cation binding site angle distributions in methanol are given in Appendix A.

As in the case of **L**, the MD simulations of alkali metal cation complexes with **l** in *N,N*-dimethylformamide indicate the formation of *trans* (Appendix A) and *cis* adducts (Appendix A) The more dominant and energetically more favorable was again the *trans* form, whereby the switching between the binding modes by internal reorientation of the included DMF molecule was observed. The portion of time in which the hydrophobic *basket* of the complexes was occupied with DMF molecules decreased with cation radius (Figure 6).

The cation coordination sphere of **l** included all ether oxygen atoms and a variable number of carbonyl oxygen atoms, and, in the case of potassium complex with **l** and rubidium complexes with both ligands, triazole nitrogen atoms (Appendix A). The binding of triazole rings was most pronounced in Rb^+^**l** complexes where one N2 atom was coordinated on average, with occasional coordination of the neighboring N3 atom (0.63 atoms in Rb**l**DMF^+^ and 0.32 atoms in Rb**l**^+^ complex on average). Similarly, as observed in MeCN and MeOH, in the Li**l**DMF^+^ adducts, the cation was coordinated by approximately one carbonyl group. The distributions of metal cation–carbonyl oxygen distances and metal cation–carbonyl oxygen–carbonyl carbon angles are given on Appendix A. As is the case with other solvents, these distributions are mostly bimodal. Peaks corresponding to the bound carbonyl groups are centered at shorter distances and at angles around 110°.

## 3. Materials and Methods

### 3.1. Materials for Synthesis and Physicochemical Investigations

All chemicals and solvents for synthesis were used without further purification and purchased from commercial sources.

The solvents, methanol (MeOH; J.T. Baker, HPLC grade), acetonitrile (MeCN; Sigma-Aldrich, St. Louis, MO, USA, HPLC grade), *N,N*-dimethylformamide (DMF; Sigma−Aldrich, HPLC grade), mQ water (H_2_O), formamide (FA; Sigma-Aldrich, spectroscopic grade), and CDCl_3_ (euriso-top, +0,03% TMS, 99,80% D) were used without further purification. The salts used for the investigation of calixarene complexation were LiClO_4_ (Sigma Aldrich, 99.99%), NaClO_4_ (Sigma Aldrich, ≥98%), KClO_4_ (Fluka, Buchs, Switzerland, ≥99,0 %), RbCl (Sigma-Aldrich, 99,8%), RbI (Sigma-Aldrich, 99,9 %), RbNO_3_ (Sigma-Aldrich, 99,7 %), CsCl (Sigma-Aldrich, 99,9%), CsCI (Sigma-Aldrich, 99,9%), CsNO_3_ (Sigma-Aldrich, 99,5%), and Cs[B(C_6_H_5_)_4_] (Sigma-Aldrich, 98%). Due to the inertness of perchlorate and tetraphenylborate anions regarding ion association, alkali salts with these anions were used in the cases when they were soluble enough.

### 3.2. Synthesis of Compound ***L***

Compound **L** was prepared according to the procedure presented in Figure 1. In a 100 mL round bottom flask, precursor **P [16]** (500 mg, 0.35 mmol) was dissolved in 50 mL of DCM. 1-Pentin (310 μL, 3.15 mmol) was then added, followed by copper(I) iodide (60 mg, 0.315 mmol), DIPEA (55 μL, 0.315 mmol), and AcOH (18 μL, 0.315 mmol). The mixture was stirred for 24 h at room temperature and, after that, DCM was evaporated. The mixture was dissolved in 150 mL of EtOAc and washed with 100 mL of 5 % NH_3_ (aq), followed by the addition of 10 × 100 mL of mQ water. The organic layer was filtered through cotton wool and evaporated. Crystallization from ethanol/water mixture yielded 520 mg (74%) of pure product.

**^1^H NMR L** (400 MHz, CDCl_3_) δ/ppm 7.37 (s, 4H), 7.25 (s, 4H), 6.78 (s, 8H), 4.88 (d, J = 12.8 Hz, 4H), 4.85 (s, 8H), 4.44 (t, J = 6.3 Hz, 8H), 4.33 (t, J = 6.3 Hz, 8H), 3.64 (t, J = 6.3 Hz, 8H), 3.47 (t, J = 6.3 Hz, 8H), 3.17 (d, J = 12.8 Hz, 4H), 2.66-2.54 (m, 16H), 1.67-1.55 (m, 16H), 1.07 (s, 36H), 0.96-0.87 (m, 24H); **^13^C NMR L** (100 MHz, CDCl_3_) δ / ppm 170.72 (s), 153.19 (s), 148.73 (s), 148.48 (s), 145.40 (s), 133.49 (s), 125.65 (d), 121.97 (d), 71.10 (t), 49.06 (t), 48.39 (t), 47.84 (t), 47.53 (t), 33.99 (s), 32.21 (t), 31.52 (q), 27.74 (t), 22.88 (t), 22.82 (t), 19.93 (q); **FTIR** (KBr, cm^−1^) 2959, 2871, 1656, 1551, 1458, 1199, 1128, 1045, 870, 572; **HRMS** (MALDI-TOF) m/z [M + H]^+^—calculated for (C_108_H_156_N_28_O_8_)—1996.2559, found 1996.2562.

### 3.3. Methods

#### 3.3.1. Microcalorimetry

Microcalorimetric measurements were performed by an isothermal titration calorimeter Microcal VP-ITC at 25.0 °C. The enthalpy changes were recorded upon stepwise, automatic addition of alkali metal salt solution (*c* = 4 × 10^−4^ mol·dm^−3^ to 0.1 mol·dm^−3^) to macrocycle solution (*c* = 0.8 – 1.5 × 10^−4^ mol·dm^−3^) or solution of alkali metal complex (*c* = 1 – 2 × 10^−4^ mol·dm^−3^) in competitive titrations. Blank experiments were carried out in order to make corrections for the enthalpy changes corresponding to the dilution of the alkali metal salt solution in the pure solvent or in the solution of another alkali metal salt (competitive experiments). The dependence of successive enthalpy change on the titrant volume was processed using the Microcal OriginPro 7.0 and OriginPro 7.5 programs in the cases of direct titrations, whereas data measured by competitive titrations were processed using HypDH program [40]. Titrations for each cation/ligand or cation/complex system were repeated three or more times.

#### 3.3.2. NMR Investigations

NMR spectra were recorded by means of a Bruker Avance III HD 400 MHz/54 mm Ascend spectrometer equipped with a 5 mm PA BBI 1H/D-BB probe head with z-gradient and automated tuning and matching accessory. All proton spectra were acquired at 25.0 °C by using 64 K data points, spectral width of 20 ppm, recycle delay of 1.0 s, and 16 scans. CDCl_3_ was used as a solvent and TMS as an internal standard for proton chemical shifts. ^1^H NMR titrations were performed by recording the spectral changes in Na**L**^+^ solutions in CDCl_3_ (*c*_0_ ≈ 6 × 10^−3^ mol·dm^−3^, *V*_0_ = 0.500 mL) upon stepwise addition of MeCN, MeOH, or DMF solutions (*c* ≈ 6 mol·dm^−3^) in CDCl_3_. The dependences of selected proton chemical shifts on the concentrations of reactants were processed using the HypNMR2008 program [41].

#### 3.3.3. Spectrophotometry

Spectrophotometric titrations were carried out at 25.0 ± 0.1 °C by means of Agilent Cary 60 spectrophotometer equipped with a thermostatting devices. The spectral changes of **L** and **l** solutions (*c*_0_ = 1.5 × 10^−4^ mol·dm^−3^, *V*_0_ = 2.0 mL) were recorded upon stepwise addition of RbNO_3_ or CsNO_3_ solutions (*c* = 2 × 10^−2^ mol·dm^−3^) into the measuring quartz cell (Hellma, Suprasil QX, *l* = 1 cm). Absorbances were sampled at 1 nm intervals, with an integration time of 0.2 s. The obtained spectrophotometric data were processed using the HypSpec program [40].

#### 3.3.4. Solubility Measurements

Saturated solutions of macrocycles **l** in methanol and **L** in acetonitrile were prepared by adding excess amounts of the solid to the solvents explored. The obtained mixtures were left in a thermostat at 25.0 °C for several days with periodical shaking in order to equilibrate. After the equilibrium had been reached, aliquots of solutions were taken for the solubility determination. The concentrations of saturated solutions of macrocycles at 25.0 °C were determined spectrophotometrically by means of an Agilent Cary 60 spectrophotometer equipped with a thermostatting device. The molar absorption coefficients of the compounds were obtained by measuring the absorbances of macrocycles’ solutions of known concentrations.

#### 3.3.5. Dissolution Enthalpies

Dissolution enthalpies of **L**, **l**, and Glc (anhydrous, Kemika, Zagreb, Croatia) in the investigated solvents were determined by means of TAM IV (TA Instruments) dissolution calorimeter at 25 °C. The samples were directly weighed into the cartridges (*V* = 20 μL), with their mass varying from 1.6 to 8.4 mg, whereas the solvent volume was constant (*V* = 17 mL), as well as the stirring rate (*v* = 60 rpm). Once equilibrium has been established, the samples were expelled into the solution and the heat flow was recorded every 5 s. The dissolution heats were corrected for the blank experiment (empty cartridge).

#### 3.3.6. Molecular Dynamics Simulations

The molecular dynamics simulations were carried out by means of the GROMACS [42,43,44,45,46,47,48] package (version 2020.5). Intramolecular and nonbonded intermolecular interactions were modelled by the CHARMM36 (Chemistry at HARvard Macromolecular Mechanics) force field [49]. Partial charges of lower rim substituent atoms of **l** were calculated with CGENFF web server [50,51,52,53]. Initial structures of free ligands were ones in which the calixarene *basket* had a conformation of a squashed *cone*. Initial structures of calixarene complexes were made by placing a cation in the center of the lower-rim cavity between ether oxygen atoms and carbonyl oxygen atoms of lower-rim substituents. The M–**L**^+^ species (M^+^ denotes alkali metal cations) were solvated in a cubical box (edge length 60 Å) of acetonitrile, methanol, or *N*,*N*-dimethylformamide, with periodic boundary conditions. Solvent boxes were equilibrated prior to solvation of calixarene ligands and corresponding complexes. Solute concentration in such a box was about 0.01 mol·dm^−3^. During the simulations of the systems, Cl^−^ ion was included to neutralize the box. The chloride counterion was held fixed at the box periphery, whereas the complex was initially positioned at the box center. In all simulations, an energy minimization procedure was performed, followed by a molecular dynamics simulation in *NpT* conditions with Berendsen barostat [54] for the duration of 5 ns. Afterwards, a 50.5 ns of *NpT* production simulation with Parrinello–Rahman barostat followed [55,56], with the time constant of 1 ps. The pressure was kept at 1 bar on average. The first 0.5 ns of production simulation were discarded upon the data analysis. The integrator used for the propagation, and also for the temperature control, was stochastic dynamics algorithm [57] with a time step of 1 fs. Temperature was kept at 298 K during simulation. The cutoff radius for nonbonded van der Waals and short-range Coulomb interactions was 15 Å. Long-range Coulomb interactions were treated by the Ewald method, as implemented in the particle mesh Ewald (PME) procedure [58]. Average molecular structures of calixarene–cation complexes were obtained by principle component analysis (PCA) on a coordination matrix whose rows contained distances between metal cation and ether and carbonyl oxygen atoms and, in some cases, cation–triazole nitrogen atoms’ distances during simulation. Angles between metal cations and carbonyl groups were added to the coordination matrix as well. The chosen structures were closest to the centroids of the most populous clusters in space defined by the first two principal components. The coordination matrix of free ligands was constructed of distances between phenol and carbonyl oxygen atoms and geometric center of phenol oxygen atoms. The angles between this geometric center and carbonyl groups were also used. Figures of molecular structures were created using VMD software [59].

## 4. Conclusions

Two important conclusions can be drawn from the comparative study of **L** and **l** binding abilities in methanol and *N,N*-dimethylformamide. First, the glucose subunits do not participate in cation binding. Their involvement would result in much larger differences between **L** and **l** hosting properties, which were found to be quite similar. Second, their influence on the ligand solubility is remarkable. With that respect, their ability to form hydrogen bonds with the solvent molecules seems to be of particular importance. This is evident from the differences between the **L** and **l** enthalpies of transfer from MeOH and DMF and the very low solubility of compound **l** in MeCN. Importantly, we showed that a conclusion whether a significant influence of a certain group on the compound solubility in a particular solvent can be gained by determining the dissolution enthalpies of free functionalities.

As far as the solvent influence on the alkali metal complexation thermodynamics is concerned, the complex stability was the highest in acetonitrile, followed by methanol and *N,N*-dimethylformamide. The NMR investigations revealed relatively low affinity of sodium complex with receptor **L** for explored solvents in CDCl_3_. The stability constant of Na**L**^+^ adducts decreased in the order MeCN > MeOH > DMF, which has been previously observed [7,11,15,35]. All cation hosting reactions were enthalpically controlled and, generally, the complex stabilities were similar to those corresponding to the other tertiary-amide calix[4]arene derivatives [11,60,61,62], whereby the peak affinity of receptors for sodium was noticed in all explored solvents. The standard complexation entropies were negative in all cases, except for Li^+^ hosting in MeCN. In other explored solvents, the binding of this cation was accompanied with the lowest entropy changes. This can be explained by entropically favorable desolvation of small Li^+^.

The results of MD simulations were in accord with experimentally obtained insights. The computationally collected data indicated weak involvement of triazole functionalities in K^+^ hosting in MeOH and Rb^+^ hosting in DMF and MeCN.

## Data Availability

The data presented in this study are available in Appendix A.

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
