# Peer review of "The Role of Triazole and Glucose Moieties in Alkali Metal Cation Complexation by Lower-Rim Tertiary-Amide Calix[4]arene Derivatives"

_molecules, 2022, doi:10.3390/molecules27020470_

Round 1

Reviewer 1 Report

Dear Sir

The authors presented the role of triazole and glucose moieties in alkali metal cation 2

complexation by lower-rim tertiary-amide calix[4]arene deriva- 3

tives. The work is interested and can be accepted but the following comments should be considered before production

My comments

1- Aim of the work should be stated clearly in introduction.

2- For "The solubilities of compounds L and l in MeOH and DMF were investigated with the aim of determining the standard Gibbs energies of their transfers using the relation", the following citations should be added

  • Journal of King Saud University-Science 27 (1), 2015, 54-62
  • Journal of Saudi Chemical Society 21, 2017, S128-S135

3- The current study should be compared with other related studies in literature

4- The authors must revise language of the manuscript before publication and the whole article must be adjusted based on journal style.

Author Response

“Aim of the work should be stated clearly in introduction.”

The Introduction section has been modified according to Reviewer’s suggestion, so that the aim of the work has been more clearly stated.  

"The solubilities of compounds L and l in MeOH and DMF were investigated with the aim of determining the standard Gibbs energies of their transfers using the relation", the following citations should be added

  • Journal of King Saud University-Science 27 (1), 2015, 54-62
  • Journal of Saudi Chemical Society 21, 2017, S128-S135"

With due respect, the papers proposed to be cited have very little in common with the submitted manuscript. However, we have followed the Reviewer’s suggestion and have cited the appropriate references (3 and 35) in the revised manuscript.

“The current study should be compared with other related studies in literature”

As suggested, in the Conclusion section of the revised manuscript the abilities of the solvent molecules to be included in the hydrophobic cone of the investigated compounds have been compared with those corresponding to other calixarene ligands previously described in the literature. In addition, the cation-hosting affinities of L an l have also been compared with those of other tertiary-amide derivatives.  

“The authors must revise language of the manuscript before publication and the whole article must be adjusted based on journal style.”

We have carefully read the paper, corrected all typos and revised the manuscript language. The manuscript style has been adjusted to match the template of Molecules.

Reviewer 2 Report

Comments to the Authors:

This is an interesting and overall well-written paper, this manuscript describes the binding of alkali metal cations with two tertiary-amide lower-rim calix[4]arenes under different conditions: in methanol, N,N-dimethylformamide and acetonitrile. Firstly, the glucose subunits do not participate in cation binding. Secondly, their influences on the ligand solubility are remarkable. Lastly, for the solvent influence on the alkali metal complexation thermodynamics, the complex stability is the highest in acetonitrile, followed by methanol and N,N-dimethylformamide. It will be a solid contribution to the Molecules and will certainly appeal to many of its readers. I address some of the main issues with the manuscript in the next few paragraphs. The conclusions are verified by enough and convincing data. It is recommended that this manuscript to be publish in Molecules after completing minor revision.

  1. Further improving the picture format to ensure that all pictures are positioned in the center.
  2. Figure 1, what are the reaction conditions for the synthesis of compound L? Please specifying reaction conditions.
  3. Table 1 and 2, what is the meaning of 盨E? Please correcting or adding explanation.
  4. Figure 8, please correcting the position of the diagram and the content below.
  5. The introduction part is too short. It is recommended to start the description from the classification and application of supramolecular chemistry, the following recently published important related papers should be cited: Theranostics 2019, 9, 3041; Chem. Soc. Rev. 2021, 50, 2839; Chem. Int. Ed. 2021, 60, 8115.

Author Response

“Further improving the picture format to ensure that all pictures are positioned in the center.”

“Figure 8, please correcting the position of the diagram and the content below.”

All graphical content throughout the manuscript has been adjusted to match the Journal’s style.

“Figure 1, what are the reaction conditions for the synthesis of compound L? Please specifying reaction conditions.”

We thank the Reviewer for pointing out this issue. The additional reaction conditions have been specified in the Scheme 1 of the revised manuscript.  

“Table 1 and 2, what is the meaning of 盨E? Please correcting or adding explanation”

We have corrected the mistake which appeared in the pdf version of manuscript (obviously due to the conversion from docx to pdf format). Now it reads ± SE (standard error).

“The introduction part is too short. It is recommended to start the description from the classification and application of supramolecular chemistry, the following recently published important related papers should be cited: Theranostics 2019, 9, 3041; Chem. Soc. Rev. 2021, 50, 2839; Chem. Int. Ed. 2021, 60, 8115.

With due respect, in our opinion the lengthy description/definition of supramolecular chemistry is unnecessary in the Introduction of the manuscript as it is not a review paper. However, we have modified the Introduction section and introduced two suggested references (no. 30 and 31 in the revised manuscript).